# Efficient chromatin accessibility mapping in situ by nucleosome-tethered tagmentation

**Steven Henikoff[1,2]\*, Jorja G Henikoff[1], Hatice S Kaya-Okur[1†], Kami Ahmad[1]**

[1]Basic Sciences Division Fred Hutchinson Cancer Research Center, Seattle, United States; [2]Howard Hughes Medical Institute, Seattle, United States

**Abstract** Chromatin accessibility mapping is a powerful approach to identify potential regulatory elements. A popular example is ATAC-seq, whereby Tn5 transposase inserts sequencing adapters into accessible DNA ('tagmentation'). CUT&Tag is a tagmentation-based epigenomic profiling method in which antibody tethering of Tn5 to a chromatin epitope of interest profiles specific chromatin features in small samples and single cells. Here, we show that by simply modifying the tagmentation conditions for histone H3K4me2 or H3K4me3 CUT&Tag, antibody-tethered tagmentation of accessible DNA sites is redirected to produce chromatin accessibility maps that are indistinguishable from the best ATAC-seq maps. Thus, chromatin accessibility maps can be produced in parallel with CUT&Tag maps of other epitopes with all steps from nuclei to amplified sequencing-ready libraries performed in single PCR tubes in the laboratory or on a home workbench. As H3K4 methylation is produced by transcription at promoters and enhancers, our method identifies transcription-coupled accessible regulatory sites.

**\*For correspondence:** steveh@fhcrc.org

**Present address:** †Altius Institute for Biomedical Sciences, Seattle, United States

## Introduction

Identification of DNA accessibility in the chromatin landscape has been used to infer active transcription ever since the seminal description of DNaseI hypersensitivity by Weintraub and Groudine more than 40 years ago (*Weintraub and Groudine, 1976*). Because nucleosomes occupy most of the eukaryotic chromatin landscape and regulatory elements are mostly free of nucleosomes when they are active, DNA accessibility mapping can potentially identify active regulatory elements genome-wide. Several additional strategies have been introduced to identify regulatory elements by DNA accessibility mapping, including digestion with Micrococcal Nuclease (MNase) (*Reeves, 1978*) or restriction enzymes (*Jack and Eggert, 1990*), DNA methylation (*Gottschling, 1992*), physical fragmentation (*Schwartz et al., 2005*) and transposon insertion (*Bownes, 1990*). With the advent of genome-scale mapping platforms, beginning with microarrays and later short-read DNA sequencing, mapping regulatory elements based on DNaseI hypersensitivity became routine (*Crawford et al., 2004*; *Dorschner et al., 2004*). Later innovations included FAIRE (*Giresi et al., 2007*) and Sono-Seq (*Auerbach et al., 2009*), based on physical fragmentation and differential recovery of cross-linked chromatin, and ATAC-seq (*Buenrostro et al., 2013*), based on preferential insertion of the Tn5 transposase. The speed and simplicity of ATAC-seq, in which the cut-and-paste transposition reaction inserts sequencing adapters in the most accessible genomic regions (tagmentation), has led to its widespread adoption in many laboratories for mapping presumed regulatory elements.

For all of these DNA accessibility mapping strategies, it is generally unknown what process is responsible for creating any particular accessible sites within the chromatin landscape. Furthermore accessibility is not all-or-none, with the median difference between an accessible and a non-accessible site in DNA estimated to be only ~20%, with no sites completely accessible or inaccessible in a population of cells (*Chereji et al., 2019*; *Oberbeckmann et al., 2019*). Despite these uncertainties,

**eLife digest** Cells keep their DNA tidy by wrapping it into structures called nucleosomes. Each of these structures contains a short section of DNA wound around a cluster of proteins called histones. Not only do nucleosomes keep the genetic code organized, they also control whether the proteins that can switch genes on or off have access to the DNA. When genes turn on, the nucleosomes unwrap, exposing sections of genetic code called 'gene regulatory elements'. These elements attract the proteins that help read and copy nearby genes so the cell can make new proteins. Determining which regulatory elements are exposed at any given time can provide useful information about what is happening inside a cell, but the procedure can be expensive.

The most popular way to map which regulatory elements are exposed is using a technique called Assay for Transposase-Accessible Chromatin using sequencing, or ATAC-seq for short. The 'transposase' in the acronym is an enzyme that cuts areas of DNA that are not wound around histones and prepares them for detection by DNA sequencing. Unfortunately, the data from ATAC-seq are often noisy (there are random factors that produce a signal that is detected but is not a 'real' result), so more sequencing is required to differentiate between real signal and noise, increasing the expense of ATAC-seq experiments. Furthermore, although ATAC-seq can identify unspooled sections of DNA, it cannot provide a direct connection between active genes and unwrapped DNA.

To find the link between unspooled DNA and active genes, Henikoff et al. adapted a technique called CUT&Tag. Like ATAC-seq, it also uses transposases to cut the genome, but it allows more control over where the cuts occur. When genes are switched on, the proteins reading them leave chemical marks on the histones they pass. CUT&Tag attaches a transposase to a molecule that recognizes and binds to those marks. This allowed Henikoff et al. to guide the transposases to unspooled regions of DNA bordering active genes. The maps of gene regulatory elements produced using this method were the same as the best ATAC-seq maps. And, because the transposases could only access gaps near active genes, the data provided evidence that genes switching on leads to regulatory elements in the genome unwrapping.

This new technique is simple enough that Henikoff et al. were able to perform it from home on the countertop of a laundry room. By tethering the transposases to histone marks it was possible to detect unspooled DNA that was active more efficiently than with ATAC-seq. This lowers laboratory costs by reducing the cost of DNA sequencing, and may also improve the detection of gaps between nucleosomes in single cells.

DNA accessibility mapping has successfully predicted the locations of active gene enhancers and promoters genome-wide, with excellent correspondence between methods based on very different strategies (*Karabacak Calviello et al., 2019*). This is likely because DNA accessibility mapping strategies rely on the fact that nucleosomes have evolved to repress transcription by blocking sites of pre-initiation complex formation and transcription factor binding (*Kornberg and Lorch, 2020*), and so creating and maintaining a nucleosome-depleted region (NDR) is a pre-requisite for promoter and enhancer function.

A popular alternative to DNA accessibility mapping for regulatory element identification is to map nucleosomes that border NDRs, typically by histone marks, including 'active' histone modifications, such as H3K4 methylation and H3K27 acetylation, or histone variants incorporated during transcription, such as H2A.Z and H3.3. The rationale for this mapping strategy is that the enzymes that modify histone tails and the chaperones that deposit nucleosome subunits are most active close to the sites of initiation of transcription, which typically occurs bidirectionally at both gene promoters and enhancers to produce stable mRNAs and unstable enhancer RNAs. Although the marks left behind by active transcriptional initiation 'point back' to the NDR, this cause-effect connection between the NDR and the histone marks is only by inference (*Wang et al., 2020*), and direct evidence is lacking that a histone mark is associated with an NDR.

Here, we show that a simple modification of our Cleavage Under Targets and Tagmentation (CUT&Tag) method for antibody-tethered in situ tagmentation can identify NDRs genome-wide at regulatory elements adjacent to transcription-associated histone marks in human cells. We provide

evidence that reducing the ionic concentration during tagmentation preferentially attracts Tn5 tethered to the H3K4me2 histone modification via a Protein A/G fusion to the nearby NDR, shifting the site of tagmentation from nucleosomes bordering the NDR to the NDR itself. Almost all transcription-coupled accessible sites correspond to ATAC-seq sites and vice-versa, and lie upstream of paused RNA Polymerase II (RNAPII). 'CUTAC' (Cleavage Under Targeted Accessible Chromatin) is conveniently performed in parallel with ordinary CUT&Tag, producing accessible site maps from low cell numbers with signal-to-noise as good as or better than the best ATAC-seq datasets.

## Results

### Streamlined CUT&Tag produces high-quality datasets with low cell numbers

We previously introduced CUT&RUN, a modification of Laemmli's Chromatin Immunocleavage (ChIC) method (*Schmid et al., 2004*), in which a fusion protein between Micrococcal Nuclease (MNase) and Protein A (pA-MNase) binds sites of antibodies bound to chromatin fragments in nuclei or permeabilized cells immobilized on magnetic beads. Activation of MNase with $Ca^{++}$ results in targeted cleavage, releasing the antibody-bound fragment into the supernatant for paired-end DNA sequencing. More recently, we substituted the Tn5 transposase for MNase in a modified CUT&RUN protocol, such that addition of $Mg^{++}$ results in a cut-and-paste 'tagmentation' reaction, in which sequencing adapters are integrated around sites of antibody binding (*Kaya-Okur et al., 2019*). In CUT&Tag, DNA purification is followed by PCR amplification, eliminating the end-polishing and ligation steps required for sequencing library preparation in CUT&RUN. Like CUT&RUN, CUT&Tag requires relatively little input material, and the low backgrounds permit low sequencing depths to sensitively map chromatin features.

We have developed a streamlined version of CUT&Tag that eliminates tube transfers, so that all steps can be efficiently performed in a single PCR tube (*Kaya-Okur et al., 2020*). However, we had not determined the suitability of the single-tube protocol for profiling low cell number samples. During the COVID-19 pandemic, we adapted this CUT&Tag-direct protocol for implementation with minimal equipment and space requirements that uses no toxic reagents, so that it can be performed conveniently and safely on a home workbench (*Figure 1—figure supplement 1*). To ascertain the ability of our CUT&Tag-direct protocol to produce DNA sequencing libraries at home with data quality comparable to those produced in the laboratory, we used frozen aliquots of native human K562 cell nuclei prepared in the laboratory and profiled there using the streamlined single-tube protocol. Aliquots of nuclei were thawed and serially diluted in Wash buffer from ~60,000 down to ~60 starting cells, where the average yield of nuclei was ~50%. We used antibodies to H3K4me3, which preferentially marks nucleosomes immediately downstream of active promoters, and H3K27me3, which marks nucleosomes within broad domains of polycomb-dependent silencing. Aliquots of nuclei were taken home and stored in a kitchen freezer, then thawed and diluted at home and profiled for H3K4me3 and H3K27me3. In both the laboratory and at home, we performed all steps in groups of 16 or 32 samples over the course of a single day through the post-PCR clean-up step, treating all samples the same regardless of cell numbers. Whether produced at home or in the lab, all final barcoded sample libraries underwent the same quality control, equimolar pooling, and final SPRI bead clean-up steps in the laboratory prior to DNA sequencing.

Tapestation profiles of libraries produced at home detected nucleosomal ladders down to 200 cells for H3K27me3 and nucleosomal and subnucleosomal fragments down to 2000 cells for H3K4me3 (*Figure 1A–B*). Sequenced fragments were aligned to the human genome using Bowtie2 and tracks were displayed using IGV. Similar results were obtained for both at-home and in-lab profiles for both histone modifications (*Figure 1C–D*) using pA-Tn5 produced in the laboratory, and results using commercial Protein A/Protein G-Tn5 (pAG-Tn5) were at least as good. All subsequent experiments reported here were performed at home using commercial pAG-Tn5, which provided results similar to those obtained using batches of lab-produced pA-Tn5 run in parallel.

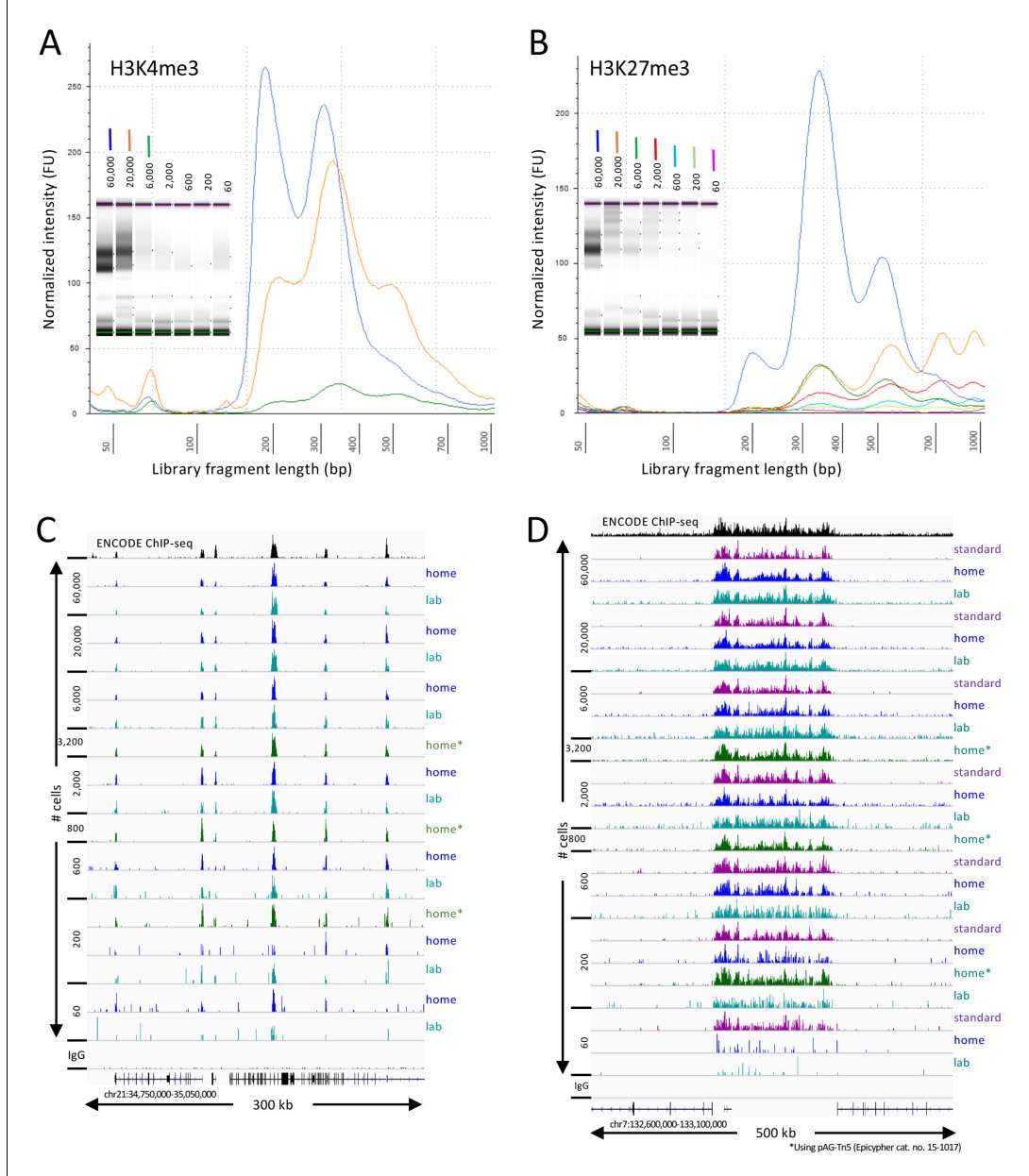

**Figure 1.** CUT&Tag-direct produces high-quality datasets on the benchtop and at home. Starting with a frozen human K562 cell aliquot, CUT&Tag-direct with amplification for 12 cycles yields detectable nucleosomal ladders for intermediate and low numbers of cells for both (**A**) H3K4me3 and (**B**) H3K27me3. The higher yield of smaller fragments with decreasing cell number suggests that reducing the total available binding sites increases the binding of antibody and/or pAG-Tn5 in limiting amounts. (**C**) Comparison of H3K4me3 CUT&Tag-direct results produced in the laboratory to those produced at home and to an ENCODE dataset (GSM733680). (**D**) Same as (**C**) for H3K27me3 comparing CUT&Tag-direct results to CUT&Tag datasets using the standard protocol (*Kaya-Okur et al., 2019*), and to an ENCODE dataset (GSM788088). pA-Tn5 was used except as indicated by asterisks for datasets produced at home using commercial pAG-Tn5 (Epicypher cat. no. 15–1017).

The online version of this article includes the following figure supplement(s) for figure 1:

**Figure supplement 1.** Equipment, supplies, reagents, and solutions for CUT&Tag on a home workbench.

## NDRs attract Tn5 tethered to nearby nucleosomes during low-salt tagmentation

Because the Tn5 domain of pA-Tn5 binds avidly to DNA, it is necessary to use elevated salt conditions to avoid tagmenting accessible DNA during CUT&Tag. High-salt buffers included 300 mM NaCl for pA-Tn5 binding, washing to remove excess protein, and tagmentation at 37°C. We have

found that other protocols based on the same principle but that do not include a high-salt wash step result in chromatin profiles that are dominated by accessible site tagmentation (*Kaya-Okur et al., 2020*).

To better understand the mechanistic basis for the salt-suppression effect, we bound pAG-Tn5 under normal high-salt CUT&Tag incubation conditions, then tagmented in low salt. We used either rapid 20-fold dilution with a prewarmed solution of 2 mM or 5 mM MgCl$_2$ or removal of the pAG-Tn5 incubation solution and addition of 50 µL 10 mM TAPS pH8.5, 5 mM MgCl$_2$. All other steps in the protocol followed our CUT&Tag-direct protocol (*Kaya-Okur et al., 2020*; *Figure 2*). Tapestation capillary gel electrophoresis of the final libraries revealed that after a 20 min incubation the effect of low-salt tagmentation on H3K4me2 CUT&Tag samples was a marked reduction in the oligo-nucleosome ladder with an increase in faster migrating fragments (*Figure 3A* and *Figure 3—figure supplement 1A–B*). CUT&Tag profiles using antibodies to most chromatin epitopes in the dilution protocol showed either little change or elevated levels of non-specific background tagmentation that obscured the targeted signal (*Figure 3—figure supplement 2*), as expected considering that we had omitted the high-salt wash step needed to remove unbound pAG-Tn5. Strikingly, under low-salt conditions, high-resolution profiles of H3K4me3 and H3K4me2 showed that the broad nucleosomal distribution of CUT&Tag around promoters for these two modifications was mostly replaced by single narrow peaks (*Figure 3B* and *Figure 3—figure supplement 3*).

To evaluate the generality of peak shifts we used MACS2 to call peaks, and plotted the occupancy over aligned peak summits. For all three H3K4 methylation marks using normal CUT&Tag high-salt tagmentation conditions we observed a bulge around the summit representing the contribution from adjacent nucleosomes on one side or the other of the peak summit (*Figure 3C*). In contrast, tagmentation under low-salt conditions revealed much narrower profiles for H3K4me3 and H3K4me2 (~40% peak width at half-height), less so for H3K4me1 (~60%), which suggests that the shift is from H3K4me-marked nucleosomes to an adjacent NDR.

To determine whether free pAG-Tn5 present during tagmentation contributes, we removed the pAG-Tn5 then added 5 mM MgCl$_2$ to tagment, and again observed narrowing of the H3K4me2 peak (*Figure 3D* 'Removal' and *Figure 3—figure supplement 1C-D*). We also observed a narrowing if we included a stringent 300 mM washing step before low-salt tagmentation (*Figure 3D*, 'Post-wash'), which indicates that peak narrowing does not require free pAG-Tn5. Inclusion of a stringent post-wash step improves consistency relative to the Dilution or Removal protocols, although it resulted in lower yields and reduced library complexity (*Figure 3—figure supplement 1E-F*). However, if a small amount of pAG-Tn5 was included during tagmentation we obtained higher yields with increased peak narrowing (*Figure 3D* 'Add-back'). Because Tn5 is inactive once it integrates its payload of adapters, and each fragment is generated by tagmentation at both ends, it is likely that a small amount of free pA(G)-Tn5 is sufficient to generate the additional small fragments where tethered pA(G)-Tn5 is limiting, albeit with higher background.

Salt ions compete with protein-DNA binding and so we suppose that tagmentation in low salt resulted in increased binding of epitope-tethered Tn5 to a nearby NDR prior to tagmentation. As H3K4 methylation is deposited in a gradient of tri- to di- to mono-methylation downstream of the +1 nucleosome from the transcriptional start site (TSS) (*Henikoff and Shilatifard, 2011*; *Soares et al., 2017*), we reasoned that the closer proximity of di- and tri-methylated nucleosomes to the NDR than mono-methylated nucleosomes resulted in preferential proximity-dependent 'capture' of Tn5. Consistent with this interpretation, we observed that the shift from broad to more peaky NDR profiles and heatmaps by H3K4me2 low-salt tagmentation was enhanced by addition of 1,6-hexanediol, a strongly polar aliphatic alcohol, and by 10% dimethylformamide, a strongly polar amide, both of which enhance chromatin accessibility (*Figure 3E–F*). NDR-focused tagmentation persisted even in the presence of both strongly polar compounds at 55°C. Enhanced localization by chromatin-disrupting conditions suggests improved access of H3K4me2-tethered Tn5 to nearby holes in the chromatin landscape during low-salt tagmentation. Localization to NDRs is more precise for small (≤120 bp) than large (>120 bp) tagmented fragments, and by resolving more closely spaced peaks inclusion of these compounds increased the number of peaks called (*Figure 3G*), also for H3K4me3-tethered Tn5 (*Figure 3—figure supplement 4*).

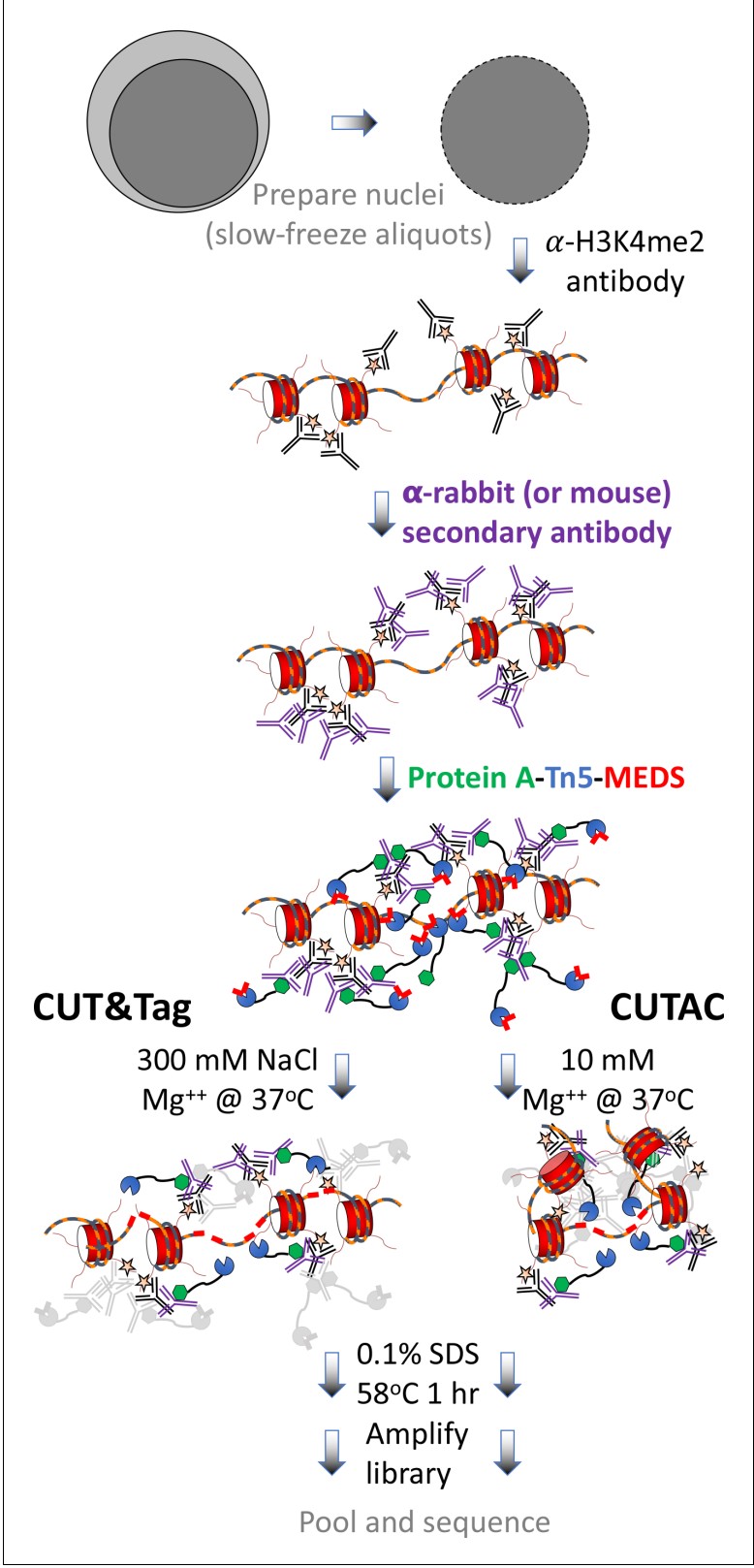

**Figure 2.** CUT&Tag with low-salt tagmentation (CUTAC). Steps in gray are lab-based and other steps were performed at home. Tagmentation can be performed by dilution, removal or post-wash. MEDS (mosaic end double-stranded annealed oligonucleotides).

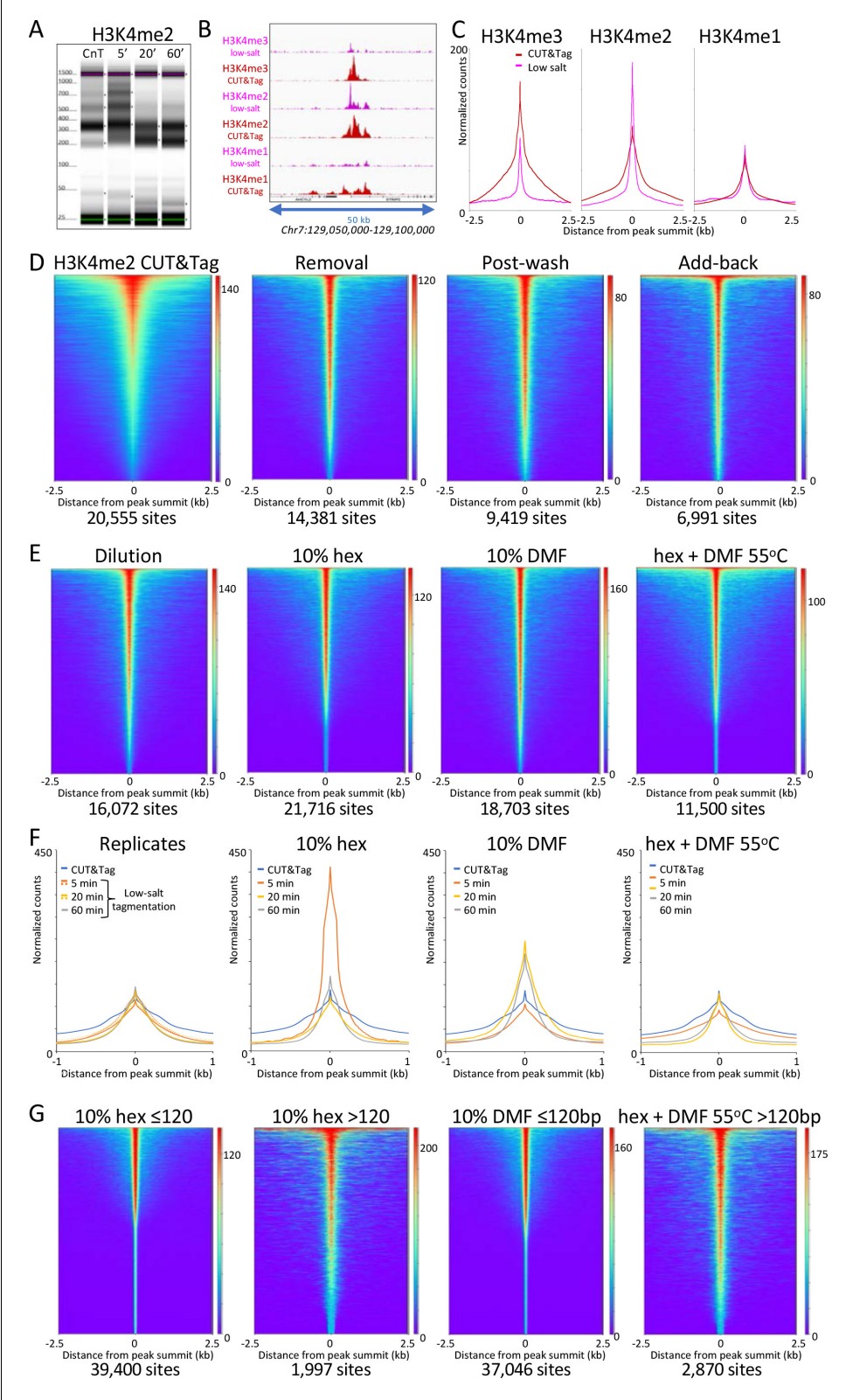

**Figure 3.** Low-salt tagmentation of H3K4me2/3 CUT&Tag samples sharpen peaks. (**A**) Tapestation gel image showing the change in size distribution from standard CUT&Tag (CnT), tagmented in the presence of 300 mM NaCl with low-salt tagmentation using the dilution protocol. (**B**) Representative tracks showing the shift observed with low-salt dilution tagmentation. (**C**) Average plots showing the narrowing of peak distributions upon low-salt tagmentation using the dilution protocol. (**D**) Heatmaps showing narrowing of H3K4me2 peaks after removing pAG-Tn5 (removal), after a stringent

*Figure 3 continued on next page*

*Figure 3 continued*

wash (post-wash), and after a stringent wash with low-salt tagmentation including a 1% pAG-Tn5 spike-in (Add-back). MACS2 was used to call peaks and heatmaps were ordered by density over the peak summits (sites). (**E**) Heatmaps showing dilution tagmentation and further narrowing of H3K4me2 peak distributions upon low-salt tagmentation (after removal) for 20 min at 37°C in the presence of 10% 1,6-hexanediol (hex) and 10% dimethylformamide (DMF) or both for 1 hr at 55°C. (**F**) Average plots showing effects of tagmentation with hex and/or DMF over time of low-salt tagmentation (after removal). (**G**) Smaller fragments (≤120 bp) dominate NDRs. Comparisons of small (≤120 bp) and large (>120 bp) fragments from CUTAC hex and DMF datasets show narrowing for small fragments around their summits. For each dataset a 3.2 million fragment random sample was split into small and large fragment groups.Removal of large fragments increases the number of peaks called (sites).

The online version of this article includes the following figure supplement(s) for figure 3:

**Figure supplement 1.** Three low-salt tagmentation protocols map chromatin hyperaccessibility.

**Figure supplement 2.** Low-salt tagmentation using various antibodies.

**Figure supplement 3.** Optimization of low-salt tagmentation conditions.

**Figure supplement 4.** H3K4me3 CUTAC shows peak narrowing and improved peak-calling with addition of 1,6-hexanediol.

## CUT&Tag low-salt tagmentation fragments coincide with ATAC-seq and DNaseI hypersensitive sites

Using CUT&Tag, we previously showed that most ATAC-seq sites are flanked by H3K4me2-marked nucleosomes in K562 cells (*Kaya-Okur et al., 2019*). However, lining up ATAC-seq datasets over peaks called using H3K4me2 CUT&Tag data resulted in smeary heatmaps, reflecting the broad distribution of peak calls over nucleosome positions flanking NDRs (*Figure 4A*). In contrast, alignment of ATAC-seq datasets over peaks called using low-salt tagmented CUT&Tag data produced narrow heatmap patterns for the vast majority of peaks (*Figure 4B*). To reflect the close similarities between fragments released by H3K4me2-tethered low-salt tagmentation as by ATAC-seq using untethered Tn5, we will refer to low-salt H3K4me2 and H3K4me3 CUT&Tag tagmentation as <u>C</u>leavage <u>U</u>nder <u>T</u>argeted <u>A</u>ccessible <u>C</u>hromatin (CUTAC).

We confirmed the similarity between CUTAC and ATAC-seq by aligning H3K4me2 CUT&Tag and CUTAC datasets over peaks called from Omni-ATAC data (*Figure 4C*). In a scatterplot comparison between CUTAC and Omni-ATAC we did not detect off-diagonal clusters that would indicate a subset of peaks found by one but not the other dataset (*Figure 4—figure supplement 1*).

To further evaluate the degree of similarity between CUTAC and ATAC-seq, we aligned the ENCODE ATAC-seq dataset over peaks called using Omni-ATAC and CUTAC, where all datasets were sampled down to 3.2 million mapped fragments with mitochondrial fragments removed. Remarkably, heatmaps produced using either Omni-ATAC or CUTAC peak calls for the same ENCODE ATAC-seq data showed occupancy of ~95% for both sets of peaks (compare right panels of *Figure 4B–C*). We found ~50% overlap between ENCODE ATAC-seq peaks and peaks called from either Omni-ATAC (50.0%) or CUTAC (51.3%) data (*Figure 4—figure supplement 2*). This equivalence between H3K4me2 CUTAC and Omni-ATAC when compared to ENCODE ATAC-seq implies that CUTAC and Omni-ATAC detect the same chromatin features. This conclusion does not hold for H3K4me3 CUTAC, because similar alignment of ENCODE ATAC-seq data resulted in only ~75% peak occupancy (*Figure 4D*) and lower correlations (*Figure 4E*), which we attribute to the greater enrichment of H3K4me3 around promoters than enhancers relative to H3K4me2.

To evaluate whether CUTAC peaks also correspond to sites of DNaseI hypersensitivity, we aligned H3K4me2 CUT&Tag and CUTAC signals over 9403 CCCTC-binding factor (CTCF) motifs scored as peaks of DNaseI sensitivity in K562 and HeLa cells. We excluded nucleosomal fragments by using only ≤120 bp fragments. We observed that 86% of the DNaseI hypersensitive CTCF sites are occupied by CUTAC signal relative to flanking regions (*Figure 4F*), which suggests equivalence of CUTAC and DNaseI hypersensitive CTCF sites. We also found that the H3K4me2 CUT&Tag sample showed detectable signal at only 53% of the CTCF sites. This improvement in detection of CTCF sites by H3K4me2 CUTAC over H3K4me2 CUT&Tag illustrates the potential of using ≤120 bp CUTAC fragment data to improve the resolution and sensitivity of transcription factor binding site motif detection.

To evaluate signal-to-noise genome-wide, we called peaks using MACS2 and calculated the <u>F</u>raction of <u>R</u>eads <u>i</u>n <u>P</u>eaks (FRiP), a data quality metric introduced by the ENCODE project (*Landt et al., 2012*). For both ENCODE ChIP-seq and our published CUT&RUN data we measured FRiP = ~0.2 for 3.2 million fragments, whereas for CUT&Tag, FRiP = ~0.4, reflecting improved signal-to-noise

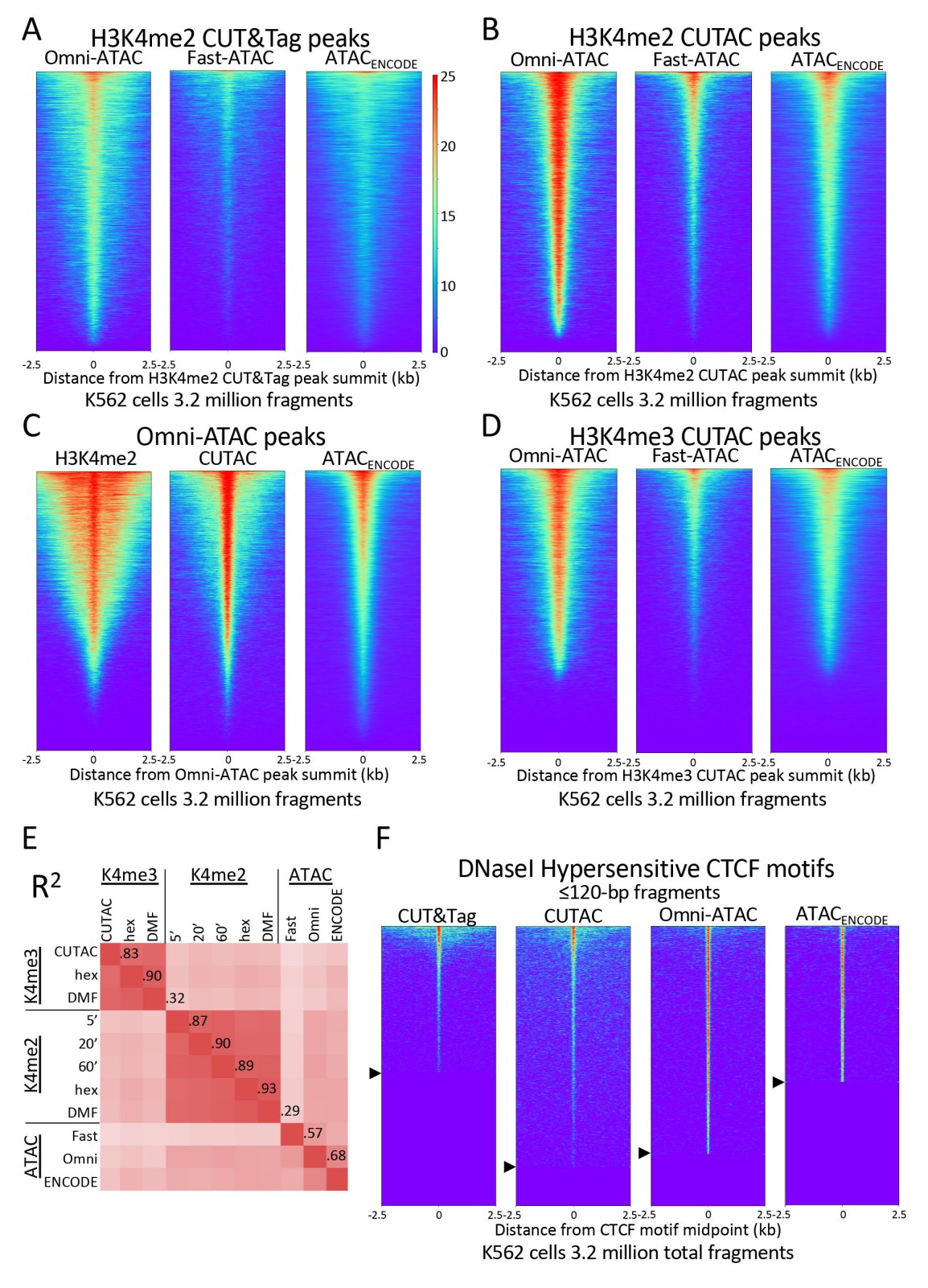

**Figure 4.** H3K4me2 CUTAC peaks correspond to ATAC-seq and DNaseI hypersensitivity peaks. (**A–D**) Heatmaps showing the correspondence between H3K4me2 CUTAC and ATAC-seq sites. Headings over each heatmap denote the source of fragments mapping to the indicated set of MACS2 peak summits, ordered by occupancy over the 5-kb interval centered over each site. CUT&Tag and CUTAC sites are from samples processed in parallel, where CUTAC tagmentation was performed by 20-fold dilution and 20 min 37°C incubation following pAG-Tn5 binding. (**E**) Correlation matrix of

*Figure 4 continued on next page*

*Figure 4 continued*

H3K4me2 and H3K4me3 CUTAC and ATAC-seq data for K562 cells. (**F**) Heatmaps showing ≤120 bp signals for H3K4me2 CUT&Tag, CUTAC and ATAC-seq at CTCF DNaseI hypersensitive sites. Arrowheads on left indicate CTCF site cutoffs.

The online version of this article includes the following figure supplement(s) for figure 4:

**Figure supplement 1.** Log$_{10}$ scatterplot of CUTAC versus Omni-ATAC (R$^2$ = 0.53).

**Figure supplement 2.** H3K4me2 CUTAC peaks correspond to ATAC-seq peaks.

relative to previous chromatin profiling methods (*Kaya-Okur et al., 2019*). Using CUT&Tag-direct, H3K4me2 CUT&Tag FRiP = 0.41 for 3.2 million fragments and ~16,000 peaks (n = 4 replicates), whereas tagmentation by dilution in 2 mM MgCl$_2$ resulted in FRiP = 0.18 for 3.2 million fragments and ~15,000 peaks (n = 4) with similar values for tagmentation by removal [FRiP = 0.21,~15,000 peaks (n = 4)]. In add-back experiments, we measured lower FRiP values after stringent washing conditions, suggesting increased background.

We also compared the number of peaks and FRiP values for CUTAC to those for ATAC-seq for K562 cells and observed that CUTAC data quality was similar to that for the Omni-ATAC method (*Corces et al., 2017*), better than ENCODE ATAC-seq (*Zhang et al., 2020*), and much better than Fast-ATAC (*Corces et al., 2016*), a previous improvement over Standard ATAC-seq (*Buenrostro et al., 2013*; *Figure 5A*). CUTAC is relatively insensitive to tagmentation times, with similar numbers of peaks and similar FRiP values for samples tagmented for 5, 20 and 60 min (*Figure 5A*). We attribute the robustness of CUT&Tag and CUTAC to the tethering of Tn5 to specific chromatin epitopes, so that when tagmentation goes to completion there is little untethered Tn5 that would increase background levels. When we measured peak numbers and FRiP values for ATAC-seq for K562 data deposited in the Gene Expression Omnibus (GEO) from multiple laboratories, we observed a wide range of data quality (*Figure 5B*, even from very recent submissions from expert groups: *Table 1* and *Figure 5—figure supplement 1*). We attribute this variability to the difficulty of avoiding background tagmention by excess free Tn5 in ATAC-seq protocols and subsequent release of non-specific nucleosomal fragments (*Swanson et al., 2020*).

If low-salt tagmentation sharpens peaks of DNA accessibility because tethering to neighboring nucleosomes increases the probability of tagmentation in small holes in the chromatin landscape, then we would expect smaller fragments to dominate CUTAC peaks. Indeed this is exactly what we observe for heatmaps (*Figure 5—figure supplement 2*), tracks (*Figure 5—figure supplement 3*), peak calls and FRiP values (*Figure 5C*). Excluding larger fragments results in better resolution yielding more peaks and higher FRIP values, both of which approach a maximum with fewer fragments. Moreover, the addition of strongly polar compounds during tagmentation provides a substantial improvement in peak calling and FRiPs (*Figure 5C*, turquoise and orange curves). Excluding large fragments did not improve ATAC-seq peak calls and FRiP values, which indicates that tethering to H3K4me2 is critical for maximum sensitivity and resolution of DNA accessibility maps.

## CUTAC maps transcription-coupled regulatory elements

H3K4me2/3 methylation marks active transcription at promoters (*Gilchrist et al., 2012*), which raises the question as to whether sites identified by CUTAC are also sites of RNAPII enrichment genome-wide. To test this possibility, we first aligned CUT&Tag and CUTAC data at annotated promoters displayed as heatmaps or average plots. CUT&Tag H3K4me2 peaks flank NDRs more downstream on either side than H3K4me3, confirmed by ENCODE ChIP-seq data to be the actual location of these marks (*Figure 6—figure supplement 1*). In contrast, CUTAC peaks are located in the NDR between flanking H3K4me2-marked chromatin (*Figure 6A*). CUTAC sites at promoter NDRs corresponded closely to promoter ATAC-seq sites, consistent with expectation for promoter NDRs. Thus, paired CUT&Tag and CUTAC samples can replace both ChIP-seq for an active promoter mark and ATAC-seq in a single experiment with identical processing, analysis and display.

To determine whether CUTAC sites are also sites of transcription initiation in general, we aligned CUT&Tag RNA Polymerase II (RNAPII) Serine-5 phosphate (RNAPIIS5P) CUT&Tag data over H3K4me2 CUT&Tag and CUTAC and Omni-ATAC peaks ordered by RNAPIIS5P peak intensity. When displayed as heatmaps or average plots, CUTAC datasets show a conspicuous shift into the NDR from flanking nucleosomes (*Figure 6B*).

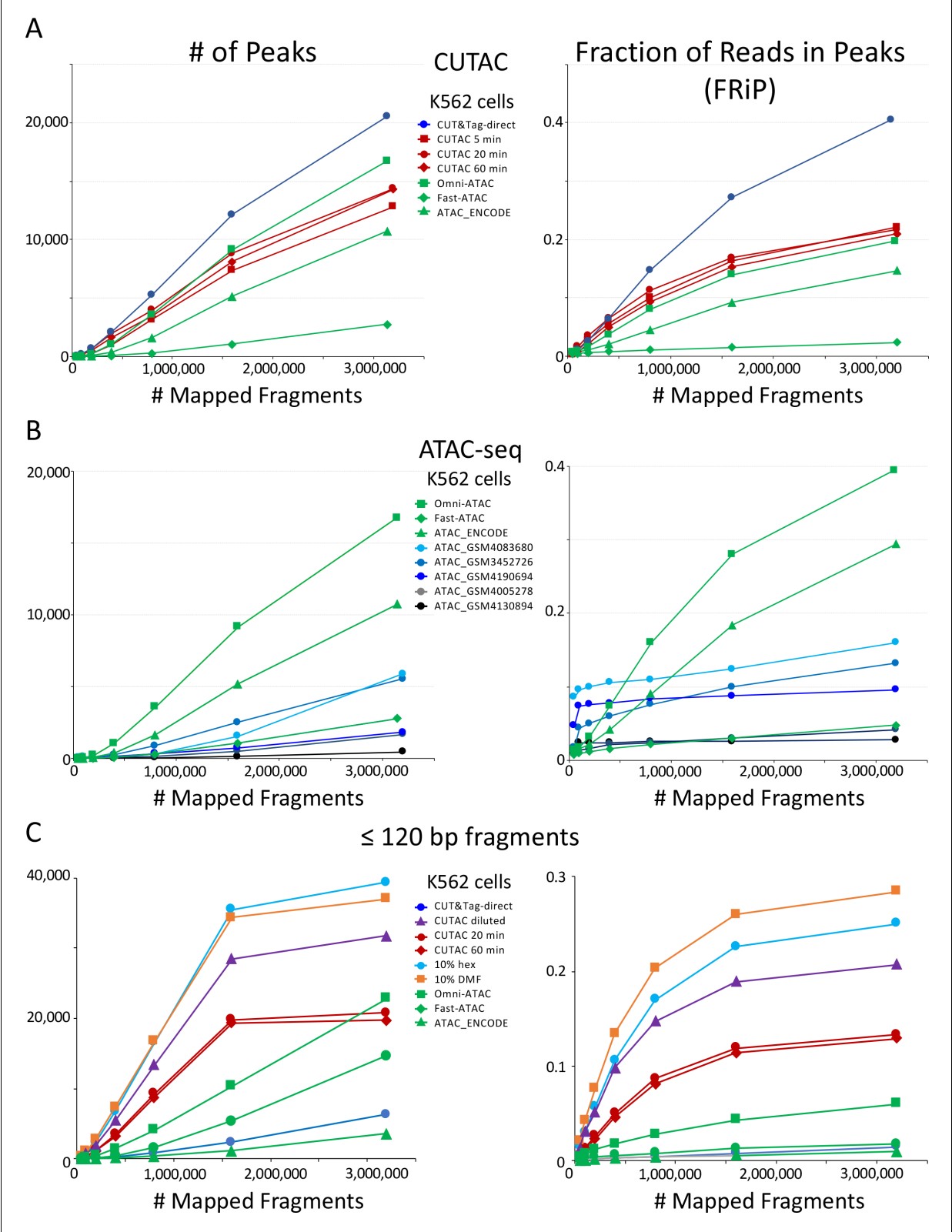

**Figure 5.** CUTAC data quality is similar to the best available ATAC-seq K562 cell data. Mapped fragments from the indicated datasets were sampled and peaks were called using MACS2. (**A**) Number of peaks (left) and fraction of reads in peaks for CUT&Tag (blue), H3K4me2 CUTAC (red) and ATAC-seq (green). Fast-ATAC is an improved version of ATAC-seq that reduces mitochondrial reads (*Corces et al., 2016*), and Omni-ATAC is an improved version that additionally improves signal-to-noise (*Corces et al., 2017*). ATAC_ENCODE is the current ENCODE standard (*Moore et al., 2020*). (**B**)

*Figure 5 continued on next page*

*Figure 5 continued*

Five other K562 ATAC-seq datasets from different laboratories were identified in GEO and mapped to hg19. MACS2 was used to call peaks. Peak numbers and FRiP values indicate a wide range of data quality found in recent ATAC-seq datasets. (C) Small H3K4me2 CUTAC fragments improve peak-calling. Hex = 1,6 hexanediol, DMF = N,N-dimethylformamide.

The online version of this article includes the following figure supplement(s) for figure 5:

**Figure supplement 1.** CUTAC data quality is similar to that of the best ATAC-seq datasets.
**Figure supplement 2.** Smaller fragments (≤120 bp) dominate NDRs.
**Figure supplement 3.** Small CUTAC fragments improve peak resolution.

Mammalian transcription also initiates at many enhancers, as shown by transcriptional run-on sequencing, which identifies sites of RNAPII pausing whether or not a stable RNA product is normally produced (*Kaikkonen et al., 2013*). Accordingly, we aligned RNAPII-profiling PRO-seq data for K562 cells over H3K4me2 CUT&Tag and CUTAC and Omni-ATAC sites, displayed as heatmaps and ordered by PRO-Seq signal intensity. The CUT&Tag sites showed broad enrichment of PRO-seq signals offset ~1 kb on either side, whereas PRO-seq signals were tightly centered around CUTAC sites, with similar results for Omni-ATAC sites (*Figure 6C*). Interestingly, alignment around TSSs, RNAPIIS5P or PRO-seq data resolved immediately flanking H3K4me2-marked nucleosomes in CUT&Tag data, which is not seen for the same data aligned on signal midpoints (*Figures 3* and *5*). Such alignment of +1 and −1 nucleosomes next to fixed NDR boundaries is consistent with nucleosome positioning based on steric exclusion (*Chereji et al., 2018*). Furthermore, the split in PRO-seq occupancies around NDRs defined by CUTAC and Omni-ATAC implies that the steady-state location of most engaged RNAPII is immediately downstream of the NDR from which it initiated. About 80% of the CUTAC sites showed enrichment of PRO-Seq signal downstream, confirming that the large majority of CUTAC sites correspond to NDRs representing transcription-coupled regulatory elements.

## Discussion

The correlation between sites of high chromatin accessibility and transcriptional regulatory elements, including enhancers and promoters, has driven the development of several distinct methods for

**Table 1.** CUTAC data quality is similar to that of the best ATAC-seq datasets.

Human K562 and H1 ES cell ATAC-seq datasets were downloaded from GEO, and Bowtie2 was used to map fragments to hg19. A sample of 3.2 million mapped fragments without Chr M was used for peak-calling by MACS2 to calculate FRiP values. Year of submission to GEO or SRA databanks is shown. % Chr M is percent of hg19-mapped fragments mapped to mitochondrial DNA.

| Sample | Source | Year | Read_type | Raw_reads | hg19-mapped | % Chr M | # Peaks | FRiP % |
|---|---|---|---|---|---|---|---|---|
| CUT&Tag-direct H1 | This study | 2020 | PE25 | 4,832,184 | 4,525,525 | 0.2 | 23,051 | 53 |
| CUT&Tag-direct K562 | This study | 2020 | PE25 | 3,252,490 | 3,144,253 | 2 | 20,555 | 41 |
| CUTAC H1 | This study | 2020 | PE25 | 2,770,901 | 2,734,092 | 1 | 16,848 | 25 |
| CUTAC K562 | This study | 2020 | PE25 | 5,973,063 | 4,785,931 | 3 | 14,381 | 22 |
| Omni-ATAC K562 SRR5657531-2 | Stanford | 2017 | PE75 | 4,407,706 | 3,181,110 | 13 | 16,737 | 20 |
| ATAC H1 GSM3677783 | Fred Hutch | 2019 | PE25 | 4,504,812 | 4,157,800 | 13 | 19,517 | 16 |
| ATAC K562 ENCFF123TMX | Stanford (ENCODE) | 2020 | PE100 | 43,473,266 | 23,942,024 | 9 | 14,369 | 11 |
| ATAC K562 GSM4083680 | U. Texas-Southwestern | 2019 | SE74 | 29,193,873 | 17,612,609 | 43 | 5894 | 8 |
| ATAC K562 GSM3452726 | Cornell U. | 2018 | PE36 | 86,907,625 | 83,038,866 | 24 | 1555 | 6.2 |
| ATAC K562 GSM4190694 | Keio U. | 2020 | PE60 | 15,363,855 | 14,067,803 | 79 | 1837 | 4.8 |
| ATAC H1 GSM4130883 | Stanford | 2020 | PE100 | 43,784,188 | 19,562,219 | 60 | 4289 | 3.2 |
| Fast-ATAC K562 SRR5657533-4 | Stanford | 2017 | PE75 | 6,702,558 | 4,677,843 | 8 | 2780 | 2.4 |
| ATAC K562 GSM4005278 | Penn State Hershey | 2020 | PE100 | 12,772,997 | 8,541,005 | 25 | 1691 | 2.1 |
| ATAC K562 GSM4130894 | Stanford | 2020 | PE100 | 45,122,834 | 19,021,462 | 86 | 449 | 1.4 |

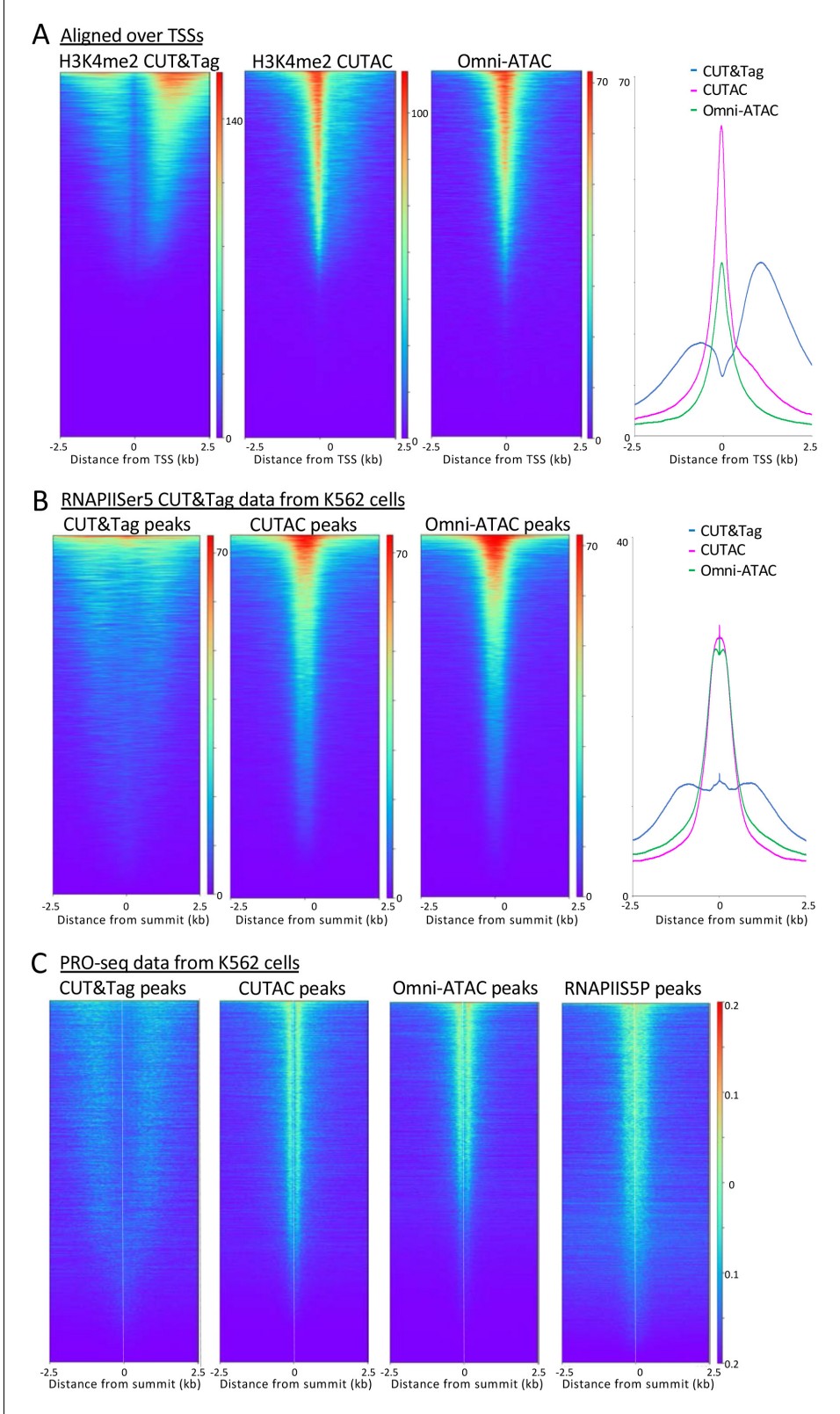

**Figure 6.** H3K4me2 CUTAC sites are coupled to transcription. (**A**) H3K4me2 fragments shift from flanking nucleosomes to the NDR upon low-salt tagmentation, corresponding closely to ATAC-seq sites. (**B**) The Serine-5 phosphate-marked initiation form of RNAPII is highly abundant over most H3K4me2 CUT&Tag, CUTAC and ATAC-seq peaks. (**C**) Run-on transcription initiates from most sites corresponding to CUTAC and ATAC-seq peaks. Both plus and minus strand PRO-seq datasets downloaded from GEO (GSM3452725) were pooled and aligned over peaks called using 3.2 million

*Figure 6 continued on next page*

*Figure 6 continued*

fragments sampled from H3K4me2 CUT&Tag, CUTAC and Omni-ATAC datasets, and also from pooled CUT&Tag replicate datasets for K562 RNA Polymerase II Serine-5 phosphate.

The online version of this article includes the following figure supplement(s) for figure 6:

**Figure supplement 1.** ChIP-seq confirms CUT&Tag localization for H3K4me2/3 flanking active promoters.

genome-wide mapping of DNA accessibility for nearly two decades (*Klein and Hainer, 2020*). However, the processes that are responsible for creating gaps in the nucleosome landscape are not completely understood. In part this uncertainty is attributable to variations in nucleosome positioning within a population of mammalian cells such that there is only a ~20% median difference in absolute DNA accessibility between DNaseI hypersensitive sites and non-hypersensitive sites genome-wide (*Chereji et al., 2019*). This suggests that DNA accessibility is not the primary determinant of gene regulation, and contradicts the popular characterization of accessible DNA sites as 'open' and the lack of accessibility as 'closed'. Moreover, there are multiple dynamic processes that can result in nucleosome depletion, including transcription, nucleosome remodeling, transcription factor binding, and replication, so that the identification of a presumed regulatory element by chromatin accessibility mapping leaves open the question as to how accessibility is established and maintained. Our CUTAC mapping method now provides a physical link between a transcription-coupled process and DNA hyperaccessibility by showing that anchoring of Tn5 to a nucleosome mark laid down by transcriptional events immediately downstream identifies presumed gene regulatory elements that are indistinguishable from those identified by ATAC-seq. The equivalence of CUTAC and ATAC at both enhancers and promoters provides support for the hypothesis that these regulatory elements are characterized by the same regulatory architecture (*Andersson et al., 2015*; *Arnold et al., 2019*).

The mechanistic basis for asserting that H3K4 methylation is a transcription-coupled event is well-established (*Henikoff and Shilatifard, 2011*; *Soares et al., 2017*). In all eukaryotes, H3K4 methylation is catalyzed by COMPASS/SET1 and related enzyme complexes, which associate with the C-terminal domain (CTD) of the large subunit of RNAPII when Serine-5 of the tandemly repetitive heptad repeat of the CTD is phosphorylated following transcription initiation. The enrichment of dimethylated and trimethylated forms of H3K4 is thought to be the result of exposure of the H3 tail to COMPASS/SET1 during RNAPII stalling just downstream of the TSS, so that these modifications are coupled to the onset of transcription (*Soares et al., 2017*). Therefore, our demonstration that Tn5 tethered to H3K4me2 or H3K4me3 histone tail residues efficiently tagments accessible sites, implies that accessibility at regulatory elements is created by events immediately following transcription initiation. This mechanistic interpretation is supported by the mapping of CUTAC sites just upstream of RNAPII, and is consistent with the recent demonstration that PRO-seq data can be used to accurately impute 'active' histone modifications (*Wang et al., 2020*). Thus CUTAC identifies active promoters and enhancers that produce enhancer RNAs, which might help explain why ~95% of ATAC-seq peaks are detected by CUTAC and vice-versa (*Figure 4B–C*).

CUTAC also provides practical advantages over other chromatin accessibility mapping methods. Like CUT&Tag-direct, all steps from frozen nuclei to purified sequencing-ready libraries for the data presented here were performed in a day in single PCR tubes on a home workbench. As it requires only a simple modification of one step in the CUT&Tag protocol, CUTAC can be performed in parallel with an H3K4me2 CUT&Tag positive control and other antibodies using multiple aliquots from each population of cells to be profiled. We have shown that three distinct protocol modifications, dilution, removal and post-wash tagmentation yield high-quality results, providing flexibility that might be important for adapting CUTAC to nuclei from diverse cell types and tissues.

Although a CUT&Tag-direct experiment requires a day to perform, and ATAC-seq can be performed in a few hours, this disadvantage of CUTAC is offset by the better control of data quality with CUTAC as is evident from the large variation in ATAC-seq data quality between laboratories (*Table 1*). In contrast, CUT&Tag is highly reproducible using native or lightly cross-linked cells or nuclei (*Kaya-Okur et al., 2020*), and as shown here H3K4me2 CUTAC maps regulatory elements with sensitivity and signal-to-noise comparable to the best ATAC-seq datasets, even better when larger fragments are computationally excluded. Although datasets from H3K4me2 CUT&Tag have lower background than datasets from CUTAC run in parallel, the combination of the two provides

both highest data quality (CUT&Tag) and precise mapping (CUTAC) using the same H3K4me2 antibody. Therefore, we anticipate that current CUT&Tag users and others will find the CUTAC option to be an attractive alternative to other DNA accessibility mapping methods for identifying transcription-coupled regulatory elements.

# Materials and methods

## Key resources table

| Reagent type (species) or resource | Designation | Source or reference | Identifiers | Additional information |
|---|---|---|---|---|
| Cell line (Human) | K562 | ATCC | Cat#CCL-243; RRID:CVCL_0004 | |
| Cell line (Human) | H1 embryonic stem cells | WiCell | Cat#WA01-lot#WB35186; RRID:CVCL_9771 | |
| Antibody | rabbit polyclonal anti-NPAT | Thermo Fisher Scientific | PA5-66839; RRID:AB_2663287 | Concentration: 1:100 |
| Antibody | guinea pig polyclonal anti-rabbit IgG | Antibodies Online | Cat#ABIN101961; RRID:AB_10775589 | Concentration: 1:100 |
| Antibody | rabbit polyclonal anti-mouse IgG | Abcam | Cat#46540; RRID:AB_2614925 | Concentration: 1:100 |
| Antibody | rabbit monoclonal anti-H3K27me3 | Cell Signaling | Cat#9733; RRID:AB_2616029 | Concentration: 1:100 |
| Antibody | rabbit polyclonal anti-H3K4me2 | Upstate | Cat#07–730-lot#3229364; RRID:AB_11213050 | Concentration: 1:100 |
| Antibody | rabbit monoclonal anti-H3K27ac | Millipore | Cat#MABE647 | Concentration: 1:100 |
| Antibody | rabbit polyclonal anti-H3K4me3 | Active Motif | Cat#39159; RRID:AB_2561020 | Concentration: 1:100 |
| Antibody | rabbit monoclonal anti-H3K4me2 | Epicypher | Cat#13–0027 | Concentration: 1:100 |
| Antibody | rabbit monoclonal anti-H3K4me1 | Epicypher | Cat#13–0026 | Concentration: 1:100 |
| Antibody | rabbit polyclonal anti-H3K9me3 | Abcam | Cat#ab8898; RRID:AB_306848 | Concentration: 1:100 |
| Antibody | rabbit monoclonal anti-H3K36me3 | Epicypher | Cat#13–0031 | Concentration: 1:100 |
| Peptide, recombinant protein | Protein A-Tn5 | Henikoff lab | doi:10.17504/protocols.io.8yrhxv6 | Concentration: 1:200 |
| Peptide, recombinant protein | Protein AG-Tn5 | Epicypher | 1 Cat#5–1117 | Concentration: 1:20-1:60 |

## Biological materials

Human K562 cells were purchased from ATCC (CCL-243) and cultured following the supplier's protocol. H1 ES cells were obtained from WiCell (WA01-lot#WB35186) and cultured following NIH 4D Nucleome guidelines. All tested negative for mycoplasma contamination using a MycoProbe kit.

## CUT&Tag-direct and CUTAC

Log-phase human K562 or H1 embryonic stem cells were harvested and prepared for nuclei in a hypotonic buffer with 0.1% Triton-X100 essentially as described (Skene and Henikoff, 2017). A detailed, step-by-step nuclei preparation protocol can be found at protocols.io.

CUT&Tag-direct was performed as described (*Kaya-Okur et al., 2020*), except that all CUTAC experiments were done on a home laundry room counter (*Figure 1—figure supplement 1*) with 32 samples run in parallel mostly over the course of a single ~8 hour day. A detailed step-by-step protocol including the three CUTAC options used in this study can be found at protocols.io. Briefly, nuclei were thawed, mixed with activated Concanavalin A beads and magnetized to remove the liquid with a pipettor and resuspended in Wash buffer (20 mM HEPES pH 7.5, 150 mM NaCl, 0.5 mM spermidine and Roche EDTA-free protease inhibitor). After successive incubations with primary antibody (1–2 hr) and secondary antibody (0.5–1 hr) in Wash buffer, the beads were washed and resuspended in pA(G)-Tn5 at 12.5 nM in 300-Wash buffer (Wash buffer containing 300 mM NaCl) for 1 hr. Incubations were performed at room temperature either in bulk or in volumes of 25–50 μL in low-retention PCR tubes. For CUT&Tag, tagmentation was performed for 1 hr in 300-Wash buffer supplemented with 10 mM $MgCl_2$ in a 50 μL volume. For CUTAC, tagmentation was performed in low-salt buffer with varying components, volumes and temperatures as described for each experiment in the figure legends. In 'dilution' tagmentation, tubes containing 25 μL of pA(G)-Tn5 incubation solution and 2 mM or 5 mM $MgCl_2$ solutions were preheated to 37°C. Tagmentation solution (475 μL) was rapidly added to the tubes and incubated for times and temperatures as indicated. In 'removal' tagmentation, tubes were magnetized, liquid was removed, and 50 μL of ice-cold 10 mM TAPS pH 8.5, 5 mM $MgCl_2$ was added, followed by incubation for times and temperatures as indicated. The 'post-wash' protocol is identical to the CUT&Tag-direct protocol except that tagmentation was performed in 10 mM TAPS pH 8.5, 5 mM $MgCl_2$ at 37°C as indicated. In 'add-back' tagmentation, the post-wash protocol was used with 10 mM TAPS pH 8.5, 5 mM $MgCl_2$ supplemented with pA(G)-Tn5 and incubated at 37°C as indicated.

Following tagmentation, CUT&Tag and CUTAC samples were chilled and magnetized, liquid was removed, and beads were washed in 50 μL 10 mM TAPS pH 8.5, 0.2 mM EDTA then resuspended in 5 μL 0.1% SDS, 10 μL TAPS pH 8.5. Following incubation at 58°C, SDS was neutralized with 15 μL of 0.67% Triton-X100, and 2 μL of 10 mM indexed P5 and P7 primer solutions were added. Tubes were chilled and 25 μL of NEBNext 2x Master mix was added and vortexed. Gap-filling and 12 cycles of PCR were performed using an MJ PTC-200 Thermocycler. Clean-up was performed by addition of 65 μL SPRI bead slurry following the manufacturer's instructions, eluted with 20 μL 1 mM Tris-HCl pH 8, 0.1 mM EDTA and 2 μL was used for Agilent 4200 Tapestation analysis. The barcoded libraries were mixed to achieve equimolar representation as desired aiming for a final concentration as recommended by the manufacturer for sequencing on an Illumina HiSeq 2500 2-lane Turbo flow cell.

## Data processing and analysis

For datasets from GEO with fragment read lengths $\geq$60 bp we ran cutadapt 2.9 with parameters -q 20 -a AGATCGGAAGAGC -A AGATCGGAAGAGC. Paired-end reads were aligned to hg19 using Bowtie2 version 2.3.4.3 with options: `–end-to-end –very-sensitive –no-unal –no-mixed – no-discordant –phred33` -I 10 - X 700. Tracks were made as bedgraph files of normalized counts, which are the fraction of total counts at each basepair scaled by the size of the hg19 genome. Peaks were called using MACS2 version 2.2.6 callpeak -f BEDPE -g hs -p le-5 –keep-dup all –SPMR. Heatmaps were produced using deepTools 3.3.1.

To produce the scatterplot (*Figure 4—figure supplement 1*) and correlation matrix (*Figure 4E*), we first removed fragments overlapping any repeat-masked region in hg19, then sampled 3.2 million fragments from each of the 11 datasets and called peaks on the merged data using MACS2. As previously described (*Meers et al., 2019*), we used a CUTAC IgG negative control, summing normalized counts within peaks and removing peaks above a threshold of the 99th percentile of normalized count sums (46,561 final peaks).

A detailed step-by-step Data Processing and Analysis Tutorial can be found at protocols.io.

## Acknowledgements

We thank Terri Bryson, Christine Codomo for sample processing, the Fred Hutch Genomics Shared Resource for DNA sequencing, members of our laboratory for helpful discussions and Paul Talbert for critically reading the manuscript. SH is an Investigator of the Howard Hughes Medical Institute. This work was supported by the Howard Hughes Medical Institute (SH), grants R01 HG010492 (SH)

and R01 GM108699 (KA) from the National Institutes of Health, and an HCA Seed Network grant from the Chan-Zuckerberg Initiative (SH).

## Additional information

### Competing interests

Steven Henikoff, Hatice S Kaya-Okur: has filed patent applications related to this work. The other authors declare that no competing interests exist.

### Funding

| Funder | Grant reference number | Author |
|---|---|---|
| National Institutes of Health | R01 HG010492 | Steven Henikoff |
| National Institutes of Health | R01 GM108699 | Kami Ahmad |
| Chan Zuckerberg Initiative | Fred Hutch HCA Seed Network | Steven Henikoff Kami Ahmad |
| Howard Hughes Medical Institute | Henikoff | Steven Henikoff |

The funders had no role in study design, data collection and interpretation, or the decision to submit the work for publication.

### Author contributions

Steven Henikoff, Conceptualization, Resources, Formal analysis, Funding acquisition, Validation, Investigation, Methodology, Writing - original draft, Writing - review and editing; Jorja G Henikoff, Data curation, Software, Formal analysis, Writing - review and editing; Hatice S Kaya-Okur, Investigation, Methodology, Writing - review and editing; Kami Ahmad, Funding acquisition, Validation, Investigation, Methodology, Writing - review and editing

### Author ORCIDs

Steven Henikoff (iD) https://orcid.org/0000-0002-7621-8685

### Decision letter and Author response

Decision letter https://doi.org/10.7554/eLife.63274.sa1
Author response https://doi.org/10.7554/eLife.63274.sa2

## Additional files

### Supplementary files

• Supplementary file 1. MSExcel spreadsheets of metadata information for each figure panel and track (Tab 1), for each dataset in GEO (Tab 2), and for other GEO/SRA database files (Tab 3) used in the study.

• Transparent reporting form

### Data availability

Sequencing data have been deposited in GEO under accession code GSE158327.

The following dataset was generated:

| Author(s) | Year | Dataset title | Dataset URL | Database and Identifier |
|---|---|---|---|---|
| Henikoff S, Kaya-Okur HS, Ahmad K | 2020 | Efficient transcription-coupled chromatin accessibility mapping in situ | https://www.ncbi.nlm.nih.gov/geo/query/acc.cgi?acc=GSE158327 | NCBI Gene Expression Omnibus, GSE158327 |

The following previously published dataset was used:

| Author(s) | Year | Dataset title | Dataset URL | Database and Identifier |
|-----------|------|---------------|-------------|-------------------------|
| Kaya-Okur HS | 2019 | CUT&Tag for efficient epigenomic profiling of small samples and single cells | https://www.ncbi.nlm.nih.gov/geo/query/acc.cgi?acc=GSE124557 | NCBI Gene Expression Omnibus, GSE124557 |

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
