## [Decision Letter]

Thank you for submitting your article "Efficient transcription-coupled chromatin accessibility mapping in situ" for consideration by *eLife*. Your article has been reviewed by three peer reviewers, and the evaluation has been overseen by a Reviewing Editor and Jessica Tyler as the Senior Editor. The following individuals involved in review of your submission have agreed to reveal their identity: Charles G Danko (Reviewer #1); Junyue Cao (Reviewer #3).

The reviewers have discussed the reviews with one another and the Reviewing Editor has drafted this decision to help you prepare a revised submission.

Essential revisions

1) Reviewers #1 and #3 raise questions regarding the possibility of some accessible regions not being profiled by CUTAC and reviewer #1 suggests to analyze regions bound by CTCF to explore this possibility. Please include these or other analyses to address this point.

2) Reviewers #1 and #3 wonder about the quantitative correlation in addition to the spatial overlap shown. Please include these analyses.

3) While the experiments presented indicate a correlation between transcription and chromatin accessibility, the term "transcription-coupled" in the title implies that a causal link has been demonstrated, but this would require manipulations. We advise you to change the title to better reflect the content of the manuscript.

Additional points

We also think that several of the other points made by the reviewers might help you strengthen this manuscript and encourage you to consider addressing them if possible. The full reviews are included below.

Reviewer #1:

This paper by the Henikoff lab introduces CUTAC, a molecular tool that allows users to sequence DNA inside nucleosome depleted regions accessible to transposition by a protein A (pA)-Tn5 fusion protein. CUTAC builds on the Henikoff lab's exciting new CUT&TAG method. Unlike the CUT&TAG protocol published recently, however, this new work uses low-salt conditions during tagmentation, which appears to promote Tn5 transposition in nucleosome depleted regions adjacent to the primary antibody. The data demonstrating that CUTAC favors transposition inside of nucleosome depleted regions is compelling and clearly shown. Moreover, the new method affords a substantial improvement in the resolution for active regulatory regions compared with CUT&TAG for histone modifications, comparable to that of high-quality ATAC-seq data. Compared with ATAC-seq, there are potentially several compelling advantages of CUTAC, including reproducibility, side-by-side library prep with CUT&TAG, and the possibility of being selective about which open chromatin regions are sequenced (see below). Between these advantages and the authors' past success and broad community interest in the CUT&RUN and CUT&TAG family of methods, I am in favor of publication. I have several comments for the authors:

1) The authors' model is that the primary antibody recruits pA-Tn5 fusion, which then transposes DNA in adjacent accessible regions. However, not all nuclease accessible chromatin is marked by H3K4me2/3. Several lines of evidence suggest that, at least CTCF binding sites have a high level of DNase-I accessibility, but many lack histone modifications indicative of active enhancers/ promoters. Most of the previous work on this subject was done using DNase-I-seq, however presumably the same signal is true for ATAC-seq?! Assuming ATAC-seq shows the same signal, I am curious to know whether CUTAC data collected using K4me2/3 antibodies shows accessibility near CTCF binding sites. An easy way to get at this would be to center on CTCF sites, and break them into classes which do/ do not contain evidence of either K4me2/3 or transcription using ENCODE data. If a heatmap shows similar signal between ATAC and CUTAC at CTCF sites associated with K3me2/3, but only ATAC shows signal at CTCF sites not associated with these marks, then it implies a degree of specificity for open chromatin near the primary antibody as would be expected from the author's model.

2) Selecting which open chromatin regions to measure could be an additional, compelling advantage of CUTAC over ATAC-seq. One could imagine, for instance, using CUTAC to find open chromatin near specific kinds of transcriptional co-activators or co-repressors, Pol II, or (possibly) transcription factors. I would imagine there are a range of applications that would benefit from something like this. Is it worth saying more about this? Or do the authors think that more exploration would be required before this could be stated with any certainty (perhaps the analysis suggested in point #1, above, will help)?

3) To what extent does CUTAC recover the quantitative amount of chromatin accessibility measured by ATAC-seq? Heatmaps suggest the two are highly correlated, as would be expected. It might be useful for readers to see scatterplots that show the correlation in integrated signal near peaks. Note that I would not necessarily expect the correlation to be perfect if there is some specificity for accessible chromatin near H3K4me2/3.

4) In some parts of the text and Abstract, I came away with the impression that CUTAC involves both H3K4me2 and H3K4me3 primary antibodies in the same sample. Based on the main text, however, I think the authors are only using one of these two marks at a time. Please clarify.

5) In Figure 5, please clarify aspects of the CUTAC experiment that were explored in earlier figures were used. Was it me2 or me3? With or without hexanediol or dimethylformamide?

Reviewer #2:

In this manuscript Henikoff et al. present a modification of CUT&Tag, a method that they developed previously to profile chromatin epitopes genome-wide. Here, they show that CUT&Tag can be applied to profile transcription-couple chromatin accessibility sites by simply altering the salt concentration during tagmentation that changes the biochemical binding preferences of the pA-Tn5 transposase. The authors have creatively shown that this method can be performed at home to yield the same results as when performed in the lab, an interesting feature given the current restrictions on laboratory occupancy. The authors claim that while CUTAC takes longer to perform that ATAC-Seq, it gives better quality data than ATAC-seq based on the variation of ATAC-seq data quality between laboratories. The authors assume that all laboratories that perform ATAC-Seq are equally proficient in the technique and that variation is simply due to the technique itself and not the experimentalist. Therefore, it is unclear if CUTAC is indeed superior to ATAC-Seq. Together with the fact that CUTAC is a very minor modification of CUT&Tag, I am not convinced that it is a sufficient advance to warrant publication as a research tool in e*Life*.

Reviewer #3:

In this manuscript, Henikoff et al. developed a novel approach for in-situ mapping of transcription-coupled chromatin accessibility. Compared with conventional ATAC-seq, this method displays several unique advantages, including high sensitivity and compatibility with parallel Cut&tag profiling. I am rather enthusiastic about the release of this work. Also it is highly appreciated that the authors already uploaded the detailed protocol to protocol.io. For publication in *eLife*, this work only has several points to be clarified as shown below:

1) In Figure 1AB, CUT&Tag-direct with different starting nuclei numbers gave very different fragment size distributions. Is there a specific reason for this? How does the input nuclei/cell number affect the genome-wide signals?

2) For the divergent outputs from CUT&Tag and CUTAC, the manuscript implies that this is due to different Tn5-DNA binding affinities between low and high salt conditions. Is it also possibly due to that the high salt simply broad the space between nearby nucleosomes for more efficient tagmentation?

3) My major concern for using this approach as a substitution of ATAC-seq is that this method may introduce bias with the use of antibody linked Tn5. Are there enriched H3K4me2 signals in the peaks detected only by H3K4me2 CUTAC compared with peaks detected only by Omni-ATACseq?

4) Figure 5 is helpful for evaluating technique efficiency and qualities. However, the number of peaks per mapped fragments is affected by the input cell/nuclei number and the library's complexity. It would be great if these comparisons are made based on the same number of input nuclei. This is also helpful for comparing the efficiencies of different approaches. Also, how similar is the CUTAC dataset compared with all other ATAC-seq datasets by correlation analysis?

5) For the broad application of the technique, it would be great if the authors can compare the library preparation cost per sample between this technique and conventional ATAC-seq.

[Editors' note: further revisions were suggested prior to acceptance, as described below.]

Thank you for resubmitting your work entitled "Efficient chromatin accessibility mapping in situ by nucleosome-tethered tagmentation " for further consideration by *eLife*. Your revised article has been evaluated by Jessica Tyler (Senior Editor) and a Reviewing Editor.

The manuscript has been improved but there are some remaining issues that need to be addressed before acceptance, as outlined below:

1) Regarding H3K4me2 and ATAC-seq, your response and the revised text state "Using an interval equal to average peak width at half-height, 51.3% of CUTAC and 50.0% of Omni-ATAC sites overlap ATAC_ENCODE peaks." As this was one of the three key requests of the reviewers, the analysis supporting these numbers should be shown, perhaps as additional panel to Figure 4 (ATAC encode peaks) or as a supplementary figure.

2) In Figure 4F, the estimate of 90% coverage of these sites with CUTAC seems generous based on the heatmap. Could you add a horizontal line to clearly indicate where you believe the cutoff is? Also your response mention that you aligned Omni-seq to these sites but it's not shown in the new figure. The reviewer had specifically asked to compare H3K4me2 positive and negative in Omni-seq and CUT&tag.

3) Did you submit Table 1? I could not find it.

---

## [Author Response]

Essential revisions1) Reviewers #1 and #3 raise questions regarding the possibility of some accessible regions not being profiled by CUTAC and reviewer #1 suggests to analyze regions bound by CTCF to explore this possibility. Please include these or other analyses to address this point.

We address these points with additional analyses detailed below. For CTCF, we show that CUTAC detects ~90% of CTCF DNaseI hypersensitive sites (new Figure 4F).

2) Reviewers #1 and #3 wonder about the quantitative correlation in addition to the spatial overlap shown. Please include these analyses.

We now provide these analyses, which are described below, including additional data (new Figure 4E and Figure 4—figure supplement 1).

3) While the experiments presented indicate a correlation between transcription and chromatin accessibility, the term "transcription-coupled" in the title implies that a causal link has been demonstrated, but this would require manipulations. We advise you to change the title to better reflect the content of the manuscript.

What we meant by the title was that the *mapping* is transcription-coupled, not chromatin accessibility itself. But we agree that the title can be misinterpreted in that way, and so we have changed it to “Efficient chromatin accessibility mapping in situ by nucleosome-tethered tagmentation”.

Additional pointsWe also think that several of the other points made by the reviewers might help you strengthen this manuscript and encourage you to consider addressing them if possible. The full reviews are included below.

We thank the reviewers for their many thoughtful comments and we have addressed each of them with new data and analyses together with textual changes as requested.

Reviewer #1:This paper by the Henikoff lab introduces CUTAC, a molecular tool that allows users to sequence DNA inside nucleosome depleted regions accessible to transposition by a protein A (pA)-Tn5 fusion protein. CUTAC builds on the Henikoff lab's exciting new CUT&TAG method. Unlike the CUT&TAG protocol published recently, however, this new work uses low-salt conditions during tagmentation, which appears to promote Tn5 transposition in nucleosome depleted regions adjacent to the primary antibody. The data demonstrating that CUTAC favors transposition inside of nucleosome depleted regions is compelling and clearly shown. Moreover, the new method affords a substantial improvement in the resolution for active regulatory regions compared with CUT&TAG for histone modifications, comparable to that of high-quality ATAC-seq data. Compared with ATAC-seq, there are potentially several compelling advantages of CUTAC, including reproducibility, side-by-side library prep with CUT&TAG, and the possibility of being selective about which open chromatin regions are sequenced (see below). Between these advantages and the authors' past success and broad community interest in the CUT&RUN and CUT&TAG family of methods, I am in favor of publication. I have several comments for the authors:1) The authors' model is that the primary antibody recruits pA-Tn5 fusion, which then transposes DNA in adjacent accessible regions. However, not all nuclease accessible chromatin is marked by H3K4me2/3. Several lines of evidence suggest that, at least CTCF binding sites have a high level of DNase-I accessibility, but many lack histone modifications indicative of active enhancers/ promoters.

The evidence that I am familiar with is based on H3K4me2 and H3K4me3 ChIP-seq signal, which is smeared-out relative to DNAseI or ATAC-seq signals, and so weak positives for ATAC-seq may be below the background level for ChIP-seq. The same relationship is seen when comparing CUTAC with CUT&Tag for H3K4me2, where the only difference is in the tagmentation buffer (Figure 4C compare left and middle panels). To test whether there is a class of nuclease-accessible chromatin not detected by CUTAC we aligned H3K4me2 CUTAC and Omni-ATAC data over ATAC_ENCODE peaks. Using an interval equal to average peak width at half-height, 51.3% of CUTAC and 50.0% of Omni-ATAC sites overlap ATAC_ENCODE peaks. If there were a class of ATAC_ENCODE peaks not adjacent to H3K4me2, then these would show up as a lower percentage of overlap with CUTAC than with Omni-ATAC. In light of the moderate correlation between CUTAC and Omni-ATAC (R^2^ = 0.53, see below), it is likely that technical variation alone can account for the degree of peak overlap we observed.

Most of the previous work on this subject was done using DNase-I-seq, however presumably the same signal is true for ATAC-seq?! Assuming ATAC-seq shows the same signal, I am curious to know whether CUTAC data collected using K4me2/3 antibodies shows accessibility near CTCF binding sites. An easy way to get at this would be to center on CTCF sites, and break them into classes which do/ do not contain evidence of either K4me2/3 or transcription using ENCODE data. If a heatmap shows similar signal between ATAC and CUTAC at CTCF sites associated with K3me2/3, but only ATAC shows signal at CTCF sites not associated with these marks, then it implies a degree of specificity for open chromatin near the primary antibody as would be expected from the author's model.

We thank Reviewer 1 for suggesting this interesting analysis. To rigorously test the assertion that there are CTCF sites that show DNaseI accessibility but lack active histone marks, we used the set of 9403 19-bp CTCF motifs with a DNaseI hypersensitive site in both K562 and HeLa cells collected in an earlier study (Skene and Henikoff, 2015: DOI: 10.7554/*eLife*.09225.001), and aligned Omni-ATAC and H3K4me2 CUTAC and CUT&Tag fragments in heatmaps. We excluded nucleosomal fragments by using only ≤120 bp fragments. We observed that ~90% of the DNaseI hypersensitive CTCF sites were enriched for CUTAC signal relative to flanking regions. This suggests equivalence of the CUTAC and DNaseI hypersensitive CTCF sites, and we have added a new paragraph and new figure panel (Figure 4E) to this section. We also found that the H3K4me2 CUT&Tag sample showed detectable signal at only ~50% of the CTCF sites. This improvement in detection of CTCF sites by H3K4me2 CUTAC over H3K4me2 CUT&Tag is further evidence that the H3K4me2 mark on nucleosomes around hypersensitive sites, but is probably too weak to be detected above background in ChIP-seq experiments.

2) Selecting which open chromatin regions to measure could be an additional, compelling advantage of CUTAC over ATAC-seq. One could imagine, for instance, using CUTAC to find open chromatin near specific kinds of transcriptional co-activators or co-repressors, Pol II, or (possibly) transcription factors. I would imagine there are a range of applications that would benefit from something like this. Is it worth saying more about this? Or do the authors think that more exploration would be required before this could be stated with any certainty (perhaps the analysis suggested in point #1, above, will help)?

We agree, and now mention this possibility to close the new paragraph describing the CTCF site comparisons: “This improvement in detection of CTCF sites by H3K4me2 CUTAC over H3K4me2 CUT&Tag illustrates the potential of using ≤120-bp CUTAC fragment data to improve the resolution and sensitivity of transcription factor binding site motif detection.”

3) To what extent does CUTAC recover the quantitative amount of chromatin accessibility measured by ATAC-seq? Heatmaps suggest the two are highly correlated, as would be expected. It might be useful for readers to see scatterplots that show the correlation in integrated signal near peaks. Note that I would not necessarily expect the correlation to be perfect if there is some specificity for accessible chromatin near H3K4me2/3.

This is an excellent suggestion. Below is the requested scatterplot (R^2^ = 0.53) on a log_10_ scale to capture the 5 order-of-magnitude dynamic range (new Figure 4—figure supplement 1). We did not detect off-diagonal clusters that would indicate a subset of peaks found by one but not the other dataset. We now include a correlation matrix to more quantitatively illustrate the relationships between the H3K4me2 and H3K4me3 CUTAC and ATAC-seq datasets as requested by reviewer 3 (new Figure 4E).

4) In some parts of the text and Abstract, I came away with the impression that CUTAC involves both H3K4me2 and H3K4me3 primary antibodies in the same sample. Based on the main text, however, I think the authors are only using one of these two marks at a time. Please clarify.

Fixed (one at a time).

5) In Figure 5, please clarify aspects of the CUTAC experiment that were explored in earlier figures were used. Was it me2 or me3? With or without hexanediol or dimethylformamide?

It was H3K4me2 (fixed), but we now provide equivalent data for H3K4me3 showing the peak-narrowing effect of 1,6-hexanediol and corresponding improvements in peak-calling (new Figure 3—figure supplement 1).

Reviewer #2:In this manuscript Henikoff et al. present a modification of CUT&Tag, a method that they developed previously to profile chromatin epitopes genome-wide. Here, they show that CUT&Tag can be applied to profile transcription-couple chromatin accessibility sites by simply altering the salt concentration during tagmentation that changes the biochemical binding preferences of the pA-Tn5 transposase. The authors have creatively shown that this method can be performed at home to yield the same results as when performed in the lab, an interesting feature given the current restrictions on laboratory occupancy. The authors claim that while CUTAC takes longer to perform that ATAC-Seq, it gives better quality data than ATAC-seq based on the variation of ATAC-seq data quality between laboratories. The authors assume that all laboratories that perform ATAC-Seq are equally proficient in the technique and that variation is simply due to the technique itself and not the experimentalist. Therefore, it is unclear if CUTAC is indeed superior to ATAC-Seq. Together with the fact that CUTAC is a very minor modification of CUT&Tag, I am not convinced that it is a sufficient advance to warrant publication as a research tool in eLife.

From a user’s perspective, the fact that a minor modification converts a method for mapping specific chromatin features into one for precisely mapping accessibility in the same experiment is a major advantage. As for the question of experimentalist proficiency, we also showed that CUTAC at home outperforms ATAC-seq generated by the ENCODE project their recent *Nature* consortium publication (Moore et al., 2020). The poorest K562 ATAC-seq dataset is from another recent Nature paper (PMID: 32728247, June 2020) performed by the same lab that is listed for generating the ENCODE data (Snyder lab, Stanford). It is hard to attribute this extreme variation in ATAC-seq data quality to experimentalist skill when both the excellent and the poor datasets are from the same expert group published a month apart. To further support our assertion that nucleosome tethering of Tn5 is advantageous over free Tn5 tagmentation, our revision now includes a comparison between CUTAC, an ATAC-seq dataset from my own lab (PMID: 31253573) and one from PMID: 32728249 for human H1 ES cells. We find that CUTAC (FRiP = 0.28) is better than our own ATAC-seq dataset (FRiP = 0.16) and much better than the H1 ATAC-seq dataset from PMID: 32728249 (FRiP = 0.014). To reduce the possibility that readers will overlook our comparisons between CUTAC and ATAC-seq from expert groups, we have promoted the table reporting these statistics from part of a supplement to Table 1 in the main body of the paper.

Reviewer #3:In this manuscript, Henikoff et al. developed a novel approach for in-situ mapping of transcription-coupled chromatin accessibility. Compared with conventional ATAC-seq, this method displays several unique advantages, including high sensitivity and compatibility with parallel Cut&tag profiling. I am rather enthusiastic about the release of this work. Also it is highly appreciated that the authors already uploaded the detailed protocol to protocol.io. For publication in eLife, this work only has several points to be clarified as shown below:1) In Figure 1AB, CUT&Tag-direct with different starting nuclei numbers gave very different fragment size distributions. Is there a specific reason for this? How does the input nuclei/cell number affect the genome-wide signals?

To explain this difference, we added the following sentence to the Figure 1 legend: “The higher yield of smaller fragments with decreasing cell number suggests that reducing the total available binding sites increases the binding of antibody and/or pAG-Tn5 in limiting amounts.”

2) For the divergent outputs from CUT&Tag and CUTAC, the manuscript implies that this is due to different Tn5-DNA binding affinities between low and high salt conditions. Is it also possibly due to that the high salt simply broad the space between nearby nucleosomes for more efficient tagmentation?

No, because the CUTAC signal is centered over annotated NDRs, not over nucleosomes marked by H3K4me2. To better illustrate that the shift is due to low-salt tagmentation and not to mobility of marked nucleosomes with 300 mM NaCl, we show that the mapping of H3K4me2 by CUT&Tag is similar to mapping by ENCODE ChIP-seq where no such mobility is possible. We now make this point in the text and in the new Figure 6—figure supplement 1.

3) My major concern for using this approach as a substitution of ATAC-seq is that this method may introduce bias with the use of antibody linked Tn5. Are there enriched H3K4me2 signals in the peaks detected only by H3K4me2 CUTAC compared with peaks detected only by Omni-ATACseq?

This point was raised by reviewer 1 and clarified above, including the new analysis showing CUTAC detection of ~90% of annotated DNaseI hypersensitive CTCF sites (new Figure 4E).

4) Figure 5 is helpful for evaluating technique efficiency and qualities. However, the number of peaks per mapped fragments is affected by the input cell/nuclei number and the library's complexity. It would be great if these comparisons are made based on the same number of input nuclei. This is also helpful for comparing the efficiencies of different approaches.

This issue is addressed in the expanded Figure 3—figure supplement 1, where we have compared nuclei from 30,000 cells for CUTUC using each of the three tagmentation variations to ATAC-seq with 50,000 nuclei and show the estimated size of each CUTAC library produced.

Also, how similar is the CUTAC dataset compared with all other ATAC-seq datasets by correlation analysis?

As described in response to reviewer 1, we now include a representative scatterplot and a correlation matrix (shown above) to more quantitatively illustrate the relationships between the H3K4me2 and H3K4me3 CUTAC and ATAC-seq datasets.

5) For the broad application of the technique, it would be great if the authors can compare the library preparation cost per sample between this technique and conventional ATAC-seq.

For each CUTAC sample in a 32-sample experiment over an 8 hour day (all CUTAC@home experiments were done on this scale) we used 0.25 µL primary and 0.25 µL secondary antibody (at 1:100) and ~$10 for Epicypher pAG-Tn5 per 25 µL incubation volume, twice that for 50 µL sample volumes. Other materials, such as Concanavalin A and SPRI paramagnetic beads (~$1 per sample for both), pipette tips (~$1 per sample) and reagents increase the cost of library preparation to perhaps $15 per sample. Based on an ATAC-seq experiment from my lab (Meers et al., Mol Cell, 2019, PMID: 31253573), the cost of materials is similar, and both procedures would take about a full day if on the same scale starting with nuclei and finishing with sequencing-ready libraries (Michael Meers, personal communication). The 32-sample scale is typical for several antibodies multiplied by the number of CUT&Tag samples run in parallel with CUTAC, but is probably too high for ATAC-seq. However, the most important cost-differential is the number of sequencing reads that are required to call peaks, and this is where the number of peaks called (sensitivity) and the fraction of reads in peaks (signal-to-noise) make a big difference. For example, we sequenced to a depth of 4.5 million paired-end 25-bp reads for our published H1 ES cell ATAC-seq data (~20,000 peaks with FRiP = 0.16 for 3.2 million fragments), whereas for the H1 ES cell ATAC-seq data from PMID: 32728247 (~450 peaks with FRiP = 0.014 for 3.2 million fragments) the authors sequenced to a depth of 45 million paired-end 100-bp reads. Our in-house sequencing cost per sample was ~$25 per sample, which exceeds the estimated cost of library preparation.

[Editors' note: further revisions were suggested prior to acceptance, as described below.]

The manuscript has been improved but there are some remaining issues that need to be addressed before acceptance, as outlined below:1) Regarding H3K4me2 and ATAC-seq, your response and the revised text state "Using an interval equal to average peak width at half-height, 51.3% of CUTAC and 50.0% of Omni-ATAC sites overlap ATAC_ENCODE peaks." As this was one of the three key requests of the reviewers, the analysis supporting these numbers should be shown, perhaps as additional panel to Figure 4 (ATAC encode peaks) or as a supplementary figure.

We have added Figure 4—figure supplement 2 with heatmaps centered over the ATAC_ENCODE peaks made from each of the data files used for the Figure 4 heatmaps.

2) In Figure 4F, the estimate of 90% coverage of these sites with CUTAC seems generous based on the heatmap. Could you add a horizontal line to clearly indicate where you believe the cutoff is? Also your response mention that you aligned Omni-seq to these sites but it's not shown in the new figure. The reviewer had specifically asked to compare H3K4me2 positive and negative in Omni-seq and CUT&tag.

We have inserted arrowheads on the left of each heatmap to indicate the cutoffs. We have also added the corresponding ≤120 bp heatmaps for Omni-ATAC and ATAC_ENCODE in Figure 4F. The precise cutoffs are: H3K4me2 CUT&Tag (53%), CUTAC (86%), Omni-ATAC (82%) and ATAC_ENCODE (55%), and we have modified the article file and the response accordingly. To avoid confusion, we removed the >120 bp heatmaps, which had been excluded from this analysis, as pointed out in the text.

3) Did you submit Table 1? I could not find it.

We apologize for the inadvertent omission. Table 1 is now in the article file.